# Biosynthesis of (±)-Differolide, an Antioxidant Isolate from *Streptomyces qaidamensis* S10ᵀ

Yujie Wu [1,2,3], Wei Zhang [2,4], Kan Jiang [5], Xue Yu [2,3,4], Shiyu Wu [2], Guangxiu Liu [2,4] and Tuo Chen [1,2,*]

1 State Key Laboratory of Cryospheric Science, Northwest Institute of Eco-Environment and Resources, Chinese Academy of Sciences, Lanzhou 730000, China; wuyujie18@mails.ucas.ac.cn
2 Key Laboratory of Extreme Environmental Microbial Resources and Engineering, Lanzhou 730000, China; ziaoshen@163.com (W.Z.); yuxue@lzb.ac.cn (X.Y.); 15619010897@163.com (S.W.); liugx@lzb.ac.cn (G.L.)
3 University of Chinese Academy of Sciences, Beijing 100049, China
4 Key Laboratory of Desert and Desertification, Northwest Institute of Eco-Environment and Resources, Chinese Academy of Sciences, Lanzhou 730000, China
5 Agronomy College, Gansu Agricultural University, Lanzhou 730000, China; jiangk19@126.com
* Correspondence: chentuo@lzb.ac.cn

**Abstract:** *Streptomyces* from unexplored or underexplored environments may be an essential source of discoveries of bioactive molecules. One such example is *Streptomyces qaidamensis* S10ᵀ, which was isolated from a sand sample collected in Qaidam Basin, Qinghai Province, China. Here, we report on (±)-differolide, an antioxidant isolated from *S. qaidamensis*, and verified with scavenging experiments on 2,2-diphenyl-1-picrylhydrazyl (DPPH). The biosynthetic gene cluster responsible for synthesizing the compound was also identified using comparative genomic methods. These results provide a basis for further study of the biological activities of (±)-differolide, which also make it possible to develop as an antioxidant medicine.

**Keywords:** *Streptomyces*; (±)-differolide; antioxidant; biosynthesis

## 1. Introduction

*Streptomyces* represent a significant source of bioactive natural products (NPs), accounting for 39% of all microbial metabolites [1]. NPs isolated from *Streptomyces* are diverse and include antibiotic, anticancer, antioxidant, and other bioactivities [2–4]. Since extreme environments form severe habitats, they may give rise to unique metabolic pathways in microorganisms. For instance, the desert isolates *Streptomyces sp*. SAJ15 produces aryl polyene, terpenoid, and macrolide compounds, and the relative abundance of different biosynthetic gene clusters (BGCs) in arid *Streptomyces* differed from the non-arid counterparts [5]. Therefore, it is necessary to search in extreme environments for potential microorganisms as a source of bioactive molecules [6]. In fact, there has been extensive research focus on extreme environments such as deep marine, desert, cryosphere, volcanic, strong acids, alkalis, and polluted water and soil. For example, Hohmann and others [7] isolated a new benzoxazole antibiotic from the marine *Streptomyces sp*. NTK 937. Elsayed et al. [8] purified chaxapeptin, a novel lasso peptide with significant in vitro inhibitory activity against human lung cancer cells, which was isolated from *Streptomyces leeuwenhoekii* strain C58 collected from the Atacama Desert. Wang et al. [9] cultured an Antarctic soil-derived *Aspergillus ochraceopetaliformis* strain and isolated of new polyketides, ochraceopones with antiviral activity against influenza viruses.

Desert habitats are challenging for microorganisms, due to the scarcity of water, high UV radiation, extreme temperatures, high salinity, and the presence of inorganic oxidants [10]. Life in extreme habitats requires the presence of unique metabolites that allow microorganisms to survive. For example, *Streptomyces qaidamensis* S10ᵀ, isolated from a sand sample collected in Qaidam Basin, Qinghai Province, China, has been verified to

exhibit anti-Methicillin-resistant Staphylococcus aureus (anti-MRSA) activity [11]. Given the particular origin of *S. qaidamensis*, we hypothesize that *S. qaidamensis* can produce NPs with antioxidant activities to ensure its ecological niche.

## 2. Materials and Methods

### 2.1. Fermentation and Extract Preparation of Strain S. qaidamensis

The strain was revived in 20 mL ISP-2 medium as seed culture for fermentation. The fermentation process was started by inoculating a flask containing Gause's I medium with a 7-day broth culture of the strain *S. qaidamensis* on a rotary shaker at 200 rpm at 28 °C. After fermentation, the supernatant was collected by centrifugation at 7000 rpm for 20 min. The supernatant was extracted three times with an equal volume of ethyl acetate (EA). Subsequently, the organic solvent was collected by filtration before rotary evaporation at 40 °C. After removing the organic solvent, the product was weighed and redissolved in methanol.

### 2.2. Antioxidant Isolate from S. qaidamensis Extract

The EA extract (7 g of a brown solid) was separated using silica gel with a step gradient of N-butanol: methanol (from 400:1 to 1:1, $v/v$). Subsequently, fraction 400:1 was separated by preparative thin-layer chromatography (TLC) at the eluent ratio dichloromethane: EA of 5:1. Finally, compound 1 (71 mg) was purified using semi-preparative liquid chromatography.

Chromatographic conditions. The chromatographic column was ODS-18 (4.6 mm × 150 mm, 10 μm), the mobile phase was acetonitrile: water at 75:25 ($v/v$), isocratic elution, the detection wavelength was 254 nm with flow rate of 2 mL/min, column temperature of 35 °C and injection volume of 1 mL.

### 2.3. Structure Identification and DPPH Radical Scavenging Assay of Antioxidant

#### 2.3.1. Structure Identification

NMR data were acquired with a high-performance digital spectrometer Bruker AVANCE III HD 400 MHz at 25 °C, and X-ray single-crystal diffraction (Bruker smart Apex 2, Billerica, MA, USA).

High-resolution electrospray ionization mass spectrometry (HR-LC-MS) data were obtained using an LTQ Orbitrap Thermo Scientific (Waltham, MA, USA) MS system coupled to a Thermo Instrument HPLC system (Accela PDA detector, Accela PDA autosampler, and Accela pump, Accela, Inc., San Ramon, CA, USA). The following conditions were used: capillary voltage of 45 V, capillary temperature of 200 °C, auxiliary gas flow rate of 10−20 arbitrary units, sheath gas flow rate of 25−40 arbitrary units, spray voltage of 4.5 kV, and mass range of 100−2000 amu (maximal resolution of 30,000). For HR-LC-MS, a C18 Sunfire analytical HPLC column (5 μm, 4.6 mm × 150 mm) was used with a mobile phase consisting of 5−100% acetonitrile with 0.1% formic acid over 30 min at a flow rate of 0.5 mL/min.

#### 2.3.2. 2,2-Diphenyl-1-picrylhydrazyl Free Radical Scavenging Assay

The 2,2-diphenyl-1-picrylhydrazyl (DPPH) radical scavenging activity was carried out according to Tan et al. [12], with modifications. Different concentrations of purified antioxidant (0.5, 1, 2, and 3 mg/mL) were mixed with equal volume 0.016% ($w/v$) DPPH in 95% ($v/v$) methanol. Therefore, the final measured concentrations of samples were 0.25, 0.5, 1, and 1.5 mg/mL, respectively. The reaction mixture was left in the dark for 30 min at room temperature, and afterward, the absorbance of the mixture was taken immediately at 515 nm using a UV-Vis spectrophotometer (ThermoFisher Scientific, Waltham, MA, USA). Methanol was used as the control. The DPPH scavenging activity was calculated using the following formula:

$$\%\text{DPPH radical scavenging activity} = \frac{\text{absorbance of control} - \text{absorbance of sample}}{\text{absorbance of control}} \times 100\%$$

*2.4. Biosynthesis of the Compound*

We used antiSMASH version 6.0 [13] for the prediction of secondary metabolite biosynthesis gene cluster (SM-BGC). Single-gene deletion mutant was constructed using an in-frame method in *Escherichia coli* K-12 [14] with some modifications.

Recombinant plasmid construction. Two sequences (about 2 kb) were selected as homology arms (left and right) upstream and downstream of the gene target for deletion. PCR reactions were carried out in 50 μL of reactants containing 2.5U of TaKaRa PrimeSTAR Max DNA polymerase (San Jose, CA, USA) 100 ng genome DNA of *S. qaidamensis* as a template, 1.0 mM of each primer (Table 1), and 200 mM dNTPs. Reactions were run for 30 cycles: 98 °C for 10 s, 62 °C for 15 s, 72 °C for 1 min, plus an additional 5 min at 72 °C. PCR products were recovered on 1% agarose gel electrophoresis. Subsequently, 50 ng of plasmid pKC1139 was treated with restriction endonuclease (BamH I and Hind III from NEB Inc., Ipswich, MA, USA), and 50 ng of the left and right arms were mixed. The recombinant plasmid was constructed using ClonExpress Multis One Step Cloning Kit (Vazyme Biotec Co., Ltd., Nanjing, China).

**Table 1.** Primers used in this study.

| Primer | Sequence (5′-3′) |
| --- | --- |
| Left-for | gtaaaacgacggccagtgccaagcttcgtcatccacgcgtcgtcgaccggc |
| Left-rev | ctggaacctcctggcggccgggcgc gccagggagcgcaagctcgacagcg |
| Right-for | gcgcccggccgccaggaggttccag |
| Right-rev | aacagctatgacatgattac gaattcaccggcgagtcccccgacgggtgctg |

The recombinant plasmid was introduced into *S. qaidamensis* from *Escherichia coli* ET12567 by conjugation [15] transfer with the assistance of pUZ8002. Since pKC1139 is a thermal-sensitive plasmid, culturing the recombinant plasmid-containing *S. qaidamensis* at 37 °C for 2–3 days can incorporate the recombinant plasmid into the *Streptomyces* genome [16]. Then, the strain obtained in the previous step was cultured without adding apramycin, the plasmid sequence in the genome of the strain was lost, and a mutant strain of *S. qaidamensis* with single-gene deletion was finally obtained and named ΔC15.

The mutant ΔC15 and wild-type (WT) strains underwent the same fermentation and extraction conditions as above, and HR-LC-MS was used to detect the yield of the target product.

**3. Results**

*3.1. Structure Identification and Bioassay of Antioxidant*

The structure of compound 1 was verified as a polyketide, and the name (±)-differolide by consulting published literature [17,18] (Figure 1a). Herein, the structure of the compound was verified by a combination of X-ray crystal diffraction (Figure 1b), NMR, and HR-LC-MS (Figure 1c). The molecular formula of (±)-differolide was $C_{12}H_{12}O_4$, and the m/z of the compound was 221.0803 according to HR-LC-MS.

Compound 1 was a pale yellow solid: [1]H NMR (400 MHz, DMSO-$d_6$) δ 7.32 (d, *J* = 2.2 Hz, 2H), 6.82 (p, *J* = 3.1 Hz, 2H), 4.91–4.74 (m, 5H), 4.46 (t, *J* = 9.1 Hz, 2H), 3.89–3.75 (m, 2H), 3.55–3.39 (m, 3H), 3.15 (q, *J* = 4.6, 4.2 Hz, 2H), 2.70–2.55 (m, 1H), 2.46–2.10 (m, 6H), 1.93–1.73 (m, 4H), 1.35–1.25 (m, 1H), 1.24 (s, 2H), 0.84 (dd, *J* = 9.8, 6.6 Hz, 1H).

[13]C NMR (101 MHz, DMSO-$d_6$) δ 174.85, 169.56, 148.96, 147.31, 136.53, 135.24, 134.81, 130.09, 129.76, 127.52, 72.07, 71.19, 71.09, 68.74, 38.39, 37.07, 34.12, 30.32, 30.26, 29.89, 29.47, 29.20, 29.01, 28.13, 25.00, 22.99, 22.11.

In a previous study, (±)-differolide, the compound was proposed to be a biological regulator, with biological activity of promoting growth of *Streptomyces* aerial hyphae and spore maturation [19]. However, such bioactivity has not been supported by experimental evidence. Considering that spore maturation was associated with *Streptomyces* stress resistance, (±)-differolide may have had an antioxidant bioactivity. In this study, we

measured the scavenging of (±)-differolide to DPPH radicals with four concentration gradients. The results showed that the DPPH scavenging rate was 20.7–57.3% in the range of (±)-differolide concentration of 0.25–1.5 mg/mL, and the calculated half-inhibition concentration (IC50) was 1.24 mg/mL (Figure 1d). This suggested that the compound was definitely an antioxidant that may have the effect of enabling *S. qaidamensis* to survive in desert niche.

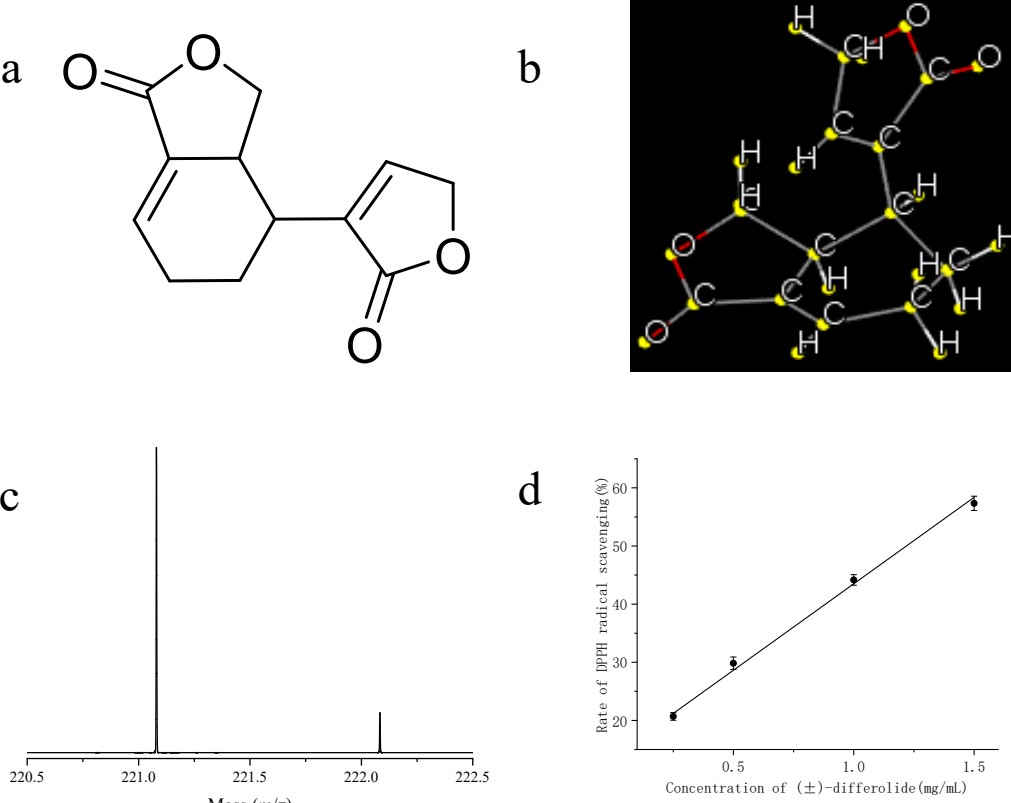

**Figure 1.** The structure and DPPH radical scavenging assay of (±)-differolide. (**a**): Structure of (±)-differolide; (**b**): X-ray crystal diffraction of (±)-differolide; (**c**): The m/z of (±)-differolide; (**d**): The rate of DPPH radical scavenging of (±)-differolide (in triplicate).

### 3.2. Biosynthesis of Differolide

The biosynthetic gene cluster of (±)-differolide was confirmed by comparing genomes between *S. qaidamensis* (CP015098.1) and *S. surantiogriseus* TÜ3149 (GCA_014649375.1), another *Streptomyces* strain capable of synthesizing (±)-differolide [20]. Genome comparison showed that eight pairs of gene clusters were similar. Predicted synthetic products of each gene cluster are shown in Table 2.

After analyzing the BGCs, we concluded that C15 BGC in *S. qaidamensis*, a PKS-type cluster, was most likely responsible for (±)-differolide biosynthesis. To test this hypothesis, the core gene 5627 in the C15 gene cluster of *S. qaidamensis* was knocked out (Figure 2a), and the ΔC15 mutant was constructed.

To explore whether the yield of (±)-differolide in the ΔC15 mutant strain is different from that of WT, both were fermented and extracted as above (in triplicate). The production of (±)-differolide was detected using HR-LC-MS, and the total ion chromatogram (TIC) spectrum of two samples is shown in Figure 2b. The diagonal line in the figure is the peak of the target compound. The results showed that the ion signal of (±)-differolide is much lower in the ΔC15 mutant strain, than that in WT. Comparison of (±)-differolide production in ΔC15 mutant and WT, suggested the yield of the compound in ΔC15 mutant was significantly lower than WT (Figure 2c), indicating that our conclusion is valid, and the C15 BGC is responsible for the biosynthesis of (±)-differolide.

**Table 2.** Genome comparison between *S. qaidamensis* S10 and *S. surantiogriseus* TÜ3149.

| S10 | TÜ3149 | Similar BGC | Type | Predicted Structure |
|---|---|---|---|---|
| C3 | C15/C23 | Melanin | Melanin | |
| C5 | C13 | Ectoine | Ectoine | |
| C8 | C22 | Desferrioxamine | Siderophore | |
| C14 | C9 | Albaflavenone | Terprne | |
| C15 | C28 | Spore pigment | T2-PKS | - |
| C22 | C10 | Hopene | Terpene | |
| C23 | C2 | Coelichelin | NRPS | |
| C25 | C12 | informatipeptin | lanthipeptide | - |

Note: T2-PKS means Type II polyketidesynthase; NRPS means non-ribosomal peptide synthase.

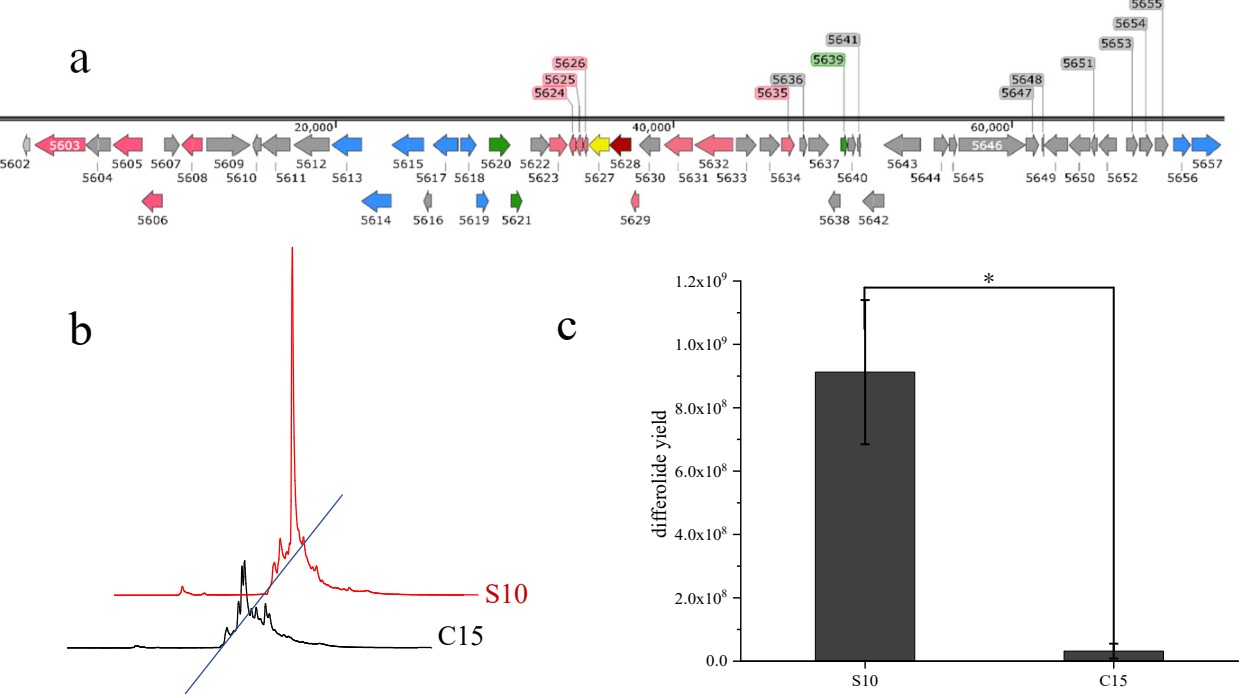

**Figure 2.** Biosynthesis of (±)-differolide. (**a**): The biosynthesis gene cluster of (±)-differolide (The function of genes was distinguished with a different color. Red: core biosynthetic genes; pink: additional biosynthetic genes; blue: transport-related genes; green: regulatory genes; grey: other genes;

yellow: gene 5627, which was knocked out). (**b**): The TIC spectrum of HR-LC-HS between ΔC15 and WT. (**c**): The yield of (±)-differolide difference between ΔC15 and WT (*: The result of significant test showed the $R^2 < 0.05$).

## 4. Discussion

In this study, we isolated a polyketide compound, (±)-differolide, from the desert-inhibiting *Streptomyces qaidamensis*. Although, (±)-differolide has been known for decades, its bioactivity has not been demonstrated before. In this study, we show its radical scavenging activity against DPPH for the first time. Researchers have been convinced for a long time that extreme *Streptomyces*-derived NPs with their structure–activity relationship (SAR) may be a highly promising source for future development of drugs against bacterial infections, especially against multidrug-resistant bacteria [6]. Many polyketides have been confirmed to have antioxidant activity. For example, rhizophols A [21], ascomindones A [22], and hexaricin F/G [23] exhibited DPPH radical scavenging activity. This indicates that (±)-differolide is precisely the antioxidant that may enable *S. qaidamensis* to survive in the niche. Considering the adaptation strategy of *Streptomyces* to different extreme conditions [24], it is possible to generate diverse biologically active compounds. These types of NPs should also attract a greater research interest. Here, we showed that it is feasible to isolate NPs with antioxidant activity from the desert *Streptomyces*.

Although the yield of (±)-differolide in *S. qaidamensis* is not very low (10 mg/L), considering the application value of its biological activity, it is necessary to synthesize the compound in large quantities. To explore the biosynthesis gene cluster of (±)-differolide, we used a comparative genomics approach in this study. The method has been successfully applied to finding biosynthetic gene clusters of target compounds in multiple strains [25,26]. It is also demonstrated here that the method enables rapid deduction of biosynthetic pathways for compounds for which no biosynthetic pathways have been reported. Biosynthesis of (±)-differolide has been verified in this study, providing a basis for its development as an antioxidant medicine.

**Author Contributions:** Conceptualization, Y.W. and T.C.; methodology, W.Z. and X.Y.; software, K.J.; validation, W.Z., G.L. and T.C.; data curation, Y.W., X.Y. and S.W.; writing—original draft preparation, Y.W.; writing—review and editing, W.Z. and T.C.; supervision, T.C.; project administration, T.C.; funding acquisition, W.Z., G.L. and T.C. All authors have read and agreed to the published version of the manuscript.

**Funding:** This research was funded by the West Light Foundation of The Chinese Academy of Sciences(xbzg-zdsys-202105), the National Key R&D Program of China (2019YFE0121100), and the Scientific Project of Gansu Province, China (20YF3WA007, 18JR2TA019, 20JR5RA548). And the APC was funded by the West Light Foundation of The Chinese Academy of Sciences(xbzg-zdsys-202105).

**Institutional Review Board Statement:** Not applicable.

**Informed Consent Statement:** Not applicable.

**Data Availability Statement:** The data presented in this study are available in the submitted article.

**Acknowledgments:** The authors are grateful to Ding Wei from Shanghai Jiao Tong University for cooperation in laboratory support, and to Zhang Qi from Fudan University for providing laboratory facilities including HR-LC-MS.

**Conflicts of Interest:** The authors declare no conflict of interest.

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
