# Peer review of "Biosynthesis of (±)-Differolide, an Antioxidant Isolate from Streptomyces qaidamensis S10T"

_applsci, doi:10.3390/app12083741_

Round 1

Reviewer 1 Report

The article titled: Biosynthesis of (±)-differolide, an antioxidant isolate from

Streptomyces qaidamensis SlO by  Wu et al., is an interesting article that has shown experimentally for the first time that differolide (a polyketide) produced by S. qaidamensis has an antioxidant function.  This is an important finding. This finding opens up the possibility of scaling up production of differolide using genetic engineering for biotechnological applications.

But the article was extremely hard to read for the following reasons:

  1. Article is not written with full scientific information. Acronyms are used without the full name.(e.g. DPPH, PKS, BGC etc.)
  2. Materials and method section does not incorporate experimental parameters, conditions and other concentration specific details. This section needs to revised massively before resubmission. What is the final concentration of DPPH after it is mixed with the respective concentrations of purified antioxidant? It is stated that the DPPH stock is 0.016% in 95% methanol. Please include details about the set up of the experiment that tests DPPH radical scavenging activity of the antioxidant.
  3. Because of the English and grammatical problem, information conveyed in lines #127-129 and lines # 146-147 are altered. Please fix these errors.
  4. In 2021 an article came out that show that Burkholderia Bacteria Produce Multiple Potentially Novel Molecules that Inhibit Carbapenem-Resistant Gram-Negative Bacterial Pathogens. This reference is not cited in the text. Here is the link to that article: Antibiotics (Basel). 2021 Feb; 10(2): 147. Published online 2021 Feb 2. doi: 10.3390/antibiotics10020147
  5. Figures should be properly labeled with concise full information
  6. Extensive editing of English language and style required

I strongly recommend the authors to edit the manuscript following the given suggestions stated above as these changes will improve the quality of the manuscript.

Reviewer 2 Report

The article is very interisting for the potential reader. is it worth publishing it after taking into account the following corrections:

-ABSTRACT: 

line 32. "The  compound was verified to be an antioxidant by scavenging experiments on DPPH", scavenging experiments in plural? in this study only uses DPPH as a antioxidant method.

25-27:  "The compound (±)-differolide has an application value as an antioxidant, and the research results in this paper make it possible to develop it into an antioxidant medicine" antioxidant activity was evaluated only by one in vitro method, so this confirmation is inconcluse, the results suggest.. only.

INTRODUCTION:

37-39 "in recent years, researchers focus on extreme environment habitats such as deep marine, desert, cryosphere, volcanic, and other strong acids, alkalis, and pollute water or soil"; should indicate the researchers of this, the references.

67: "and bioactivity assay of antioxidants", check this tittle, because is only one in vitro assay of antioxidant activity 

requiere indicate the experimental design or statistics method used for the significance of the results, mainly antioxidant activity

RESULTS

111: "The results of DPPH showed scavenging  activity of (±)-differolide was determined IC50 at 2.48 mg/mL (Fig 1d)", the description could be improved, also the points in the graph could be thinner because in a graph like that lineality is not so adequate when the ponit are so thick, also the significance of this result needs to be specified in the graph.

112: the following "The compound is precisely an antioxidant that may have the effect of enabling S.qaidamensis to protect its niche in the desert".  to make this statement requires a little more clarification, focusing on the compound, for example, the chemical structure of the compound could be related to the mechanism of the method used to evaluate antioxidant activity and that this suggests its potential use.

-Fig 1d. a correction in the x-axis for "concentration" is requiered, considering the unit on the Y-axis as the percentage of inhibition, it would be better understood by the way in which the one mentioned in the text.

-Table 1. check the type for terpene, and the strcutres on this table needs more resolution

-127: "The ΔΔC15 mutant was fermentation for three copies, together with wild type(WT) strain",  check "fermentation" or was fermented?

129-130" The total ions chromatogram (TIC) of HR-LC-HS(Fig 2b) showed that the yield of (±)-differolide in ΔC15 mutant was 2 orders of magnitude
 lower than WT(Fig 2c)" for this confirmation a significant test is requiered, also in the fig 2c. shoul describe the bars in graph meaning and indicate the significance between S10 and C15.

DISCUSSION

this section needs to be improved, reflecting the relevance of (±)-differolide as an antioxidant or bioactive compound. in general more references in this section should be included. 

clarify what is the major conclusion?

REFERENCES: 

Please add articles from "applied sciences" to your bibliography

Reviewer 3 Report

The manuscript entitled: Biosynthesis of (±)-differolide, an antioxidant isolate from Streptomyces qaidamensis S10T presents an important study regarding biosynthesis, structure identification and antioxidant activity of (±)-differolide from Streptomyces qaidamensis S10T.

Still, the present manuscripts need some improvements, as the following mentions.

First of all, it needs moderate editing of the English language, as the articles are not always properly used, some of the phrases are not being well-formed, and some sentences are not understandable.

The authors should discuss the results registered for DPPH scavenging activity. They are only mentioning the IC50 value and present the obtained graphic, but no explanations are provided.

The references are not presented in the journal requirements manner.

Round 2

Reviewer 1 Report

The authors have taken into account the reviewer's suggestions. The quality of the manuscript has improved.

Reviewer 3 Report

The authors replied to all requests. Thank you!